# Clinical Validation of the Vitro HPV Screening Assay for Its Use in Primary Cervical Cancer Screening

**DOI:** 10.3390/cancers16071322

**Published:** 2024-03-28

**Authors:** Beatriz Bellosillo, Raquel Ibáñez, Esther Roura, Laura Monfil, Laura Asensio-Puig, Isabel Álvarez, Mercè Muset, Yolanda Florencia, Sonia Paytubi, Álvaro de Andrés-Pablo, Susana Calvo, Laia Serrano-Munné, Miguel Ángel Pavón, Belen Lloveras

**Affiliations:** 1Department of Pathology, Hospital del Mar, 08003 Barcelona, Spain; bbellosillo@psmar.cat (B.B.); aamancha@sescam.jccm.es (I.Á.); mermuset@gmail.com (M.M.); scalvo@psmar.cat (S.C.); lserranomunne@psmar.cat (L.S.-M.); 2Cancer Research Program, IMIM (Hospital del Mar Medical Research Institute), 08003 Barcelona, Spain; 3Department of Medicine and Life Sciences (MELIS), University Pompeu Fabra, Doctor Aiguader 88, 08003 Barcelona, Spain; 4Cancer Epidemiology Research Programme, Catalan Institute of Oncology—Bellvitge Biomedical Research Institute (IDIBELL), 08908 L’Hospitalet de Llobregat, Spain; raquelip@iconcologia.net (R.I.); eroura@iconcologia.net (E.R.); lmonfil_ext@iconcologia.net (L.M.); lasensio@idibell.cat (L.A.-P.); yflorencia@iconcologia.net (Y.F.); spaytubic@iconcologia.net (S.P.); adeandres@idibell.cat (Á.d.A.-P.); mpavon@iconcologia.net (M.Á.P.); 5Centro de Investigación Biomédica en Red de Epidemiología y Salud Pública—CIBERESP, 28029 Madrid, Spain

**Keywords:** HPV, human papillomavirus, cervical cancer, cervical screening, genotyping, validation, Vitro HPV

## Abstract

**Simple Summary:**

The advances in cervical cancer screening have mostly focused on high-risk human papillomavirus (HR HPV) detection as the primary screening tool in women over 30 years old. Although many new HPV assays have been commercialized during the last few years, only a few of them are validated according to international guidelines. Recently, new approaches for genotype-based risk-stratification and triage of HPV positive women have been developed. In this study, the Vitro HPV Screening assay targeting 14 oncogenic genotypes (HPV16, HPV18, and another 12 HR HPV genotypes) has demonstrated an appropriate clinical performance to be used as primary screening test. Additionally, the 12 HR HPV genotypes have been individually genotyped with a complementary assay performed by reverse dot blot hybridization of the PCR product from the HPV Screening assay. In conclusion, this new assay represents a unique integrated solution for cervical cancer screening with the extended HPV genotyping as a clinical tool for risk determination.

**Abstract:**

Many scientific societies have issued guidelines to introduce population-based cervical cancer screening with HPV testing. The Vitro HPV Screening assay is a fully automatic multiplex real-time PCR test targeting the L1 GP5+/GP6+ region of HPV genome. The assay detects 14 high risk (HR) HPV genotypes, identifying individual HPV16 and HPV18 genotypes, and the HPV-positive samples for the other 12 HR HPV types are subsequently genotyped with the HPV Direct Flow Chip test. Following international guidelines, the aim of this study was to validate the clinical accuracy of the Vitro HPV Screening test on ThinPrep-collected samples for its use as primary cervical cancer screening, using as comparator the validated cobas^®^ 4800 HPV test. The non-inferiority analysis showed that the clinical sensitivity and specificity of the Vitro HPV Screening assay for a diagnosis of cervical intraepithelial neoplasia of grade 2 or worse (CIN2+) were not inferior to those of cobas^®^ 4800 HPV (*p* = 0.0049 and *p* < 0.001 respectively). The assay has demonstrated a high intra- and inter-laboratory reproducibility, also among the individual genotypes. The Vitro HPV Screening assay is valid for cervical cancer screening and it provides genotyping information on HPV-positive samples without further sample processing in a fully automated workflow.

## 1. Introduction

Cervical cancer accounts for more than 300,000 deaths worldwide each year and virtually all cases are caused by high-risk human papillomavirus (HR HPV) infection [1]. While most HR HPV infections, usually acquired at the beginning of sexual life, are cleared within months, some persist. In this case, HPV gene products disrupt control of the normal cell cycle, leading to accumulation of somatic mutations, preneoplastic lesions and, eventually, cancer [2]. However, cervical cancer is preventable. Vaccines for the most frequent HR HPV types are commercially available but worldwide coverage is still limited and screening remains one of the best strategies to prevent invasive cancer [3,4,5]. In 2018, the WHO called for a “coordinated action globally to eliminate cervical cancer”, and mathematical modelling of different strategies in different geographical regions, combining vaccination and screening, as well as appropriate treatment of women with preneoplasia or cancer, are the basis for protocols in different situations [4,6].

Since there is scientific evidence that infection with HR HPV is the main cause of cervical cancer and precursor lesions [7], the use of tests to detect viral DNA or RNA in a sample from the cervix is recommended by national and international organizations (i.e., WHO) as a primary test for cervical cancer screening, replacing cytology [8,9,10,11,12]. Many studies have shown that HPV detection has a higher sensitivity for cervical intraepithelial neoplasia grade 2 or worse (CIN2+) compared to cytology (1.52 relative sensitivity (95% Confident Interval (CI): 1.24 to 1.86)) when Hybrid capture (HC2; Qiagen, Gaithersburg, MD, USA) was compared to conventional cytology [13]. However, specificity is slightly lower in HPV tests (0.94 relative specificity (95% CI: 0.92 to 0.96)) when HC2 was compared to conventional cytology, which may lead to the detection of too many subclinical infections with no morphological lesion, and a risk of overdiagnosis, overtreatment, and psychological problems [14]. HPV detection test characteristics, including the type of technique (hybridization, signal amplification, DNA or RNA, targeted gene, or other), the HPV types detected or the sample to be analyzed, have a strong impact on both sensitivity and specificity [15,16]. Therefore, prior to its implementation, a new HPV test must be clinically validated for its use in cervical cancer screening [17]. Since the priority is the detection of disease (preferably a preneoplastic lesion that can be treated) and not just transient HPV infections, the goal of an HPV test in screening should be to reach a good “clinical sensitivity for detection of CIN2+” instead of the highest analytical sensitivity. In 2009, a group of experts published guidelines for the validation of HPV tests to be used as a primary screening test in women over 30 years old. In short, the key aspects of a new HPV test for cervical cancer screening are well-defined sensitivity and specificity, which should be not inferior to reference tests that have accumulated sufficient evidence and are considered the gold standards [18]. The reference tests considered at the time of the publication of the guidelines were HC2 and GP5+/GP6+ PCR EIA, which had been clinically validated in large prospective cohorts and randomized controlled trials [19,20,21,22,23,24]. However, some other commercially available tests have been clinically validated in the last few years and are more extensively used for different technical reasons, like automation, high throughput or availability. There are much data validating these new HPV tests published in recent years and they are now an alternative gold standard for validating new HPV tests. This is the case for the cobas^®^ 4800 HPV Test (Roche Molecular Diagnostics, Pleasanton, CA, USA), a real-time PCR test that detects 14 HR HPV in three different fluorescent channels, targeting the L1 region of the HPV genome. HPV16 and HPV18 are detected separately and the other 12 HR HPV types (31, 33, 35, 39, 45, 51, 52, 56, 58, 59, 66, and 68) are detected as a pool by a cocktail of probes in a separated channel. The cobas^®^ 4800 HPV Test has been extensively validated against standard HPV comparator assays in multiple studies [15,25] and it was evaluated against cytology in a regulatory trial that showed strong reduction in the longitudinal risk of CIN3+ among cobas^®^ HPV-negative women [26,27]. Furthermore, it has already been used as a reference comparator for the validation of other assays [17]. Finally, it was approved by the Food and Drug administration (FDA) to be used as a first-line HPV-alone test for cervical cancer screening [28].

The Vitro HPV Screening assay (Vitro S. A., Sevilla, Spain) is a fully automated HPV multiplex real-time PCR test targeting the L1 viral region GP5+/6+. It is intended to qualitatively detect 14 HR HPV genotypes, identifying HPV16 and HPV18 genotypes individually and the other 12 HR genotypes (31, 33, 35, 39, 45, 51, 52, 56, 58, 59, 66, 68) as a pool. The test reports cycle threshold (Ct) values for each channel and includes the human beta-globin gene as internal control that is co-amplified with the L1 gene of HPV genotypes. Additionally, the 12 HR HPV genotypes can be individually genotyped with a complementary assay performed by reverse dot blot hybridization of the screening PCR product (HPV Direct Flow Chip test, Vitro S. A. Sevilla, Spain).

The objective of this study was to evaluate the clinical performance as well as the reproducibility of the Vitro HPV Screening test on ThinPrep-collected samples (Hologic, Marlborough, MA, USA) from women aged 30 years and older, according to the international guidelines for primary cervical cancer screening [18], using as a gold standard assay the cobas^®^ 4800 HPV. Furthermore, the validation was complemented with extended genotyping and genotype-specific concordance analysis.

## 2. Materials and Methods

### 2.1. Sample Selection

The study was performed using residual cervical specimens from women 30 years old or older, undergoing opportunistic HPV-based cervical cancer screening in the city of Barcelona (Spain) during the period of February 2021 to February 2022. For the samples included in the sensitivity group, the period was from February 2021 to January 2022, and for the group of specificity, from December 2021 to February 2022. All samples had been collected in ThinPrep medium and previously evaluated with the cobas^®^ 4800 HPV Test, which is the routine screening test. According to the screening protocol of this area [29], only HPV-positive samples had a cytological study performed and HPV-negative women would have another HPV test after five years. In cases with positive cytology (atypical squamous cervical cells of undetermined significance (ASC-US) or worse), the patient was referred to colposcopy and a biopsy was taken in case of abnormality. Histology and cytology data collected from the pathology records were recorded in an anonymized database for the whole validation study group.

To conduct the sensitivity study, 60 consecutive samples with a histologically proven diagnosis of CIN2+ were selected. The cytological samples from these women had been collected in the previous three months, at most. For the study of specificity, 844 consecutives samples from women with either an HPV-negative result or an HPV-positive result without a diagnosis of CIN2+ or subsequent biopsy diagnostic <CIN2 were selected from the same population, in a correlative way. They were not correlative regarding HPV routine testing, but they were according to the date that the biopsy was performed.

In both groups, only samples with insufficient material for repeating the HPV tests with the Vitro HPV Screening test were discarded. All these tests were performed at the Molecular Laboratory of the Pathology Department of Hospital del Mar.

The study of intra- and inter-laboratory reproducibility was performed on 561 samples, including HR HPV-positive (>30%) and HPV-negative cases, as stipulated by the international guidelines [18]. First, the samples were processed twice at Hospital del Mar for intra-laboratory reproducibility proposes and then at the Infections and Cancer Laboratory of Cancer Epidemiology Research Program of Catalan Institute of Oncology (ICO) for the inter-laboratory analysis. For the individual genotype intra-laboratory reproducibility, the samples were re-processed twice at ICO’s laboratory.

### 2.2. Cobas^®^ 4800 HPV Test

The cobas^®^ 4800 HPV Test was used as a gold standard for this validation. It is a fully automated test based on a simultaneous extraction of the HPV and cellular DNA, followed by a real-time PCR designed to detect 14 HR HPV types using LightCyclerH480 technology. HPV16 and HPV18 are detected separately and the other 12 HR HPV types (31, 33, 35, 39, 45, 51, 52, 56, 58, 59, 66 and 68) are detected as a pool by a cocktail of probes with 3 different fluorochromes. In the same reaction the human beta-globin gene is amplified and detected by a different fluorochrome as a control for the entire process. A positive and a negative control are also included in each run. The results are expressed as negative or positive for each HPV16, HPV18 or 12 HR HPVs, using a Ct-value–cutoff of 40 to define HPV-positive. When neither beta-globin nor HPV is detected, the result is invalid. The cobas^®^ 4800 HPV Test was performed according to the manufacturer’s protocol. The interpretation of the amplification and detection stage was performed using the software supplied with the cobas^®^ 4800 platform. We used the cobas^®^ 4800 HPV Test as the reference assay, as its sensitivity, specificity, and reproducibility have been demonstrated to be equivalent and statistically non-inferior to Hybrid Capture 2 in cross-sectional comparisons [30,31].

### 2.3. Vitro HPV Screening Test

All samples were processed with the Vitro HPV Screening test using an automated solution (Figure 1). The assay is a multiplex real-time PCR containing a pool of primers targeting the L1 region of the HPV genome and 14 genotype-specific fluorescent probes. The test specifically identifies HPV types 16 and 18 in separate fluorescent channels, and the genotypes 31, 33, 35, 39, 45, 51, 52, 56, 58, 59, 66, and 68 are detected as a pool in a different channel. The human beta-globin gene is amplified and detected in a separate channel as an internal control for cellularity, sample extraction, and amplification efficiency. A positive and a negative control are also included for assay performance and to exclude contamination. Results are expressed as negative or positive for HPV16, HPV18 or 12 HR HPV. Samples were categorized as HPV-positive or HPV-negative for genotypes 16, 18, or 12 HR genotypes, based on the Ct value of 40 for the assay. The Ct value was defined following sensitivity and specificity assays developed by the manufacturer. When neither beta-globin nor HPV is detected, the result is considered invalid. The extraction of DNA from the clinical samples was performed with the MAIS automated system (Vitro S.A.) and the RNA/DNA Pathogen Extraction kit (Vitro S.A.). The automated process starts with the primary clinical sample and performs the nucleic acid extraction and the PCR plate setup. The minimum amount (ng) of cellular DNA for the test was established at 10 ng/reaction. The amplification was carried out in the Vitrocycler Real-time PCR device (Vitro S.A.) and the results were automatically analyzed with the OVTS software (V.1.0.11.1) (Vitro S.A.).

### 2.4. HPV Direct Flow CHIP Test

The HPV Direct Flow Chip assay is designed to simultaneously genotype 35 HPV types, 18 high-risk HPV types (16, 18, 26, 31, 33, 35, 39, 45, 51, 52, 53, 56, 58, 59, 66, 68, 73, and 82), and 17 low-risk HPV types (6, 11, 40, 42, 43, 44, 54, 55, 61, 62, 67, 69, 70, 71, 72, 81, and 84) by multiplex PCR and reverse dot blot hybridization with genotype-specific probes, based on DNA Flow Technology [32,33]. A DNA flow-based automatic hybriSpot™ platform allows the amplified DNA to bind to complementary probes printed into a microarray HPV CHIP in a three-dimensional porous environment. Once the binding between the specific amplicons and their corresponding probes has occurred, the signal is visualized by a colorimetric immunoenzymatic reaction with streptavidin–alkaline phosphatase and a chromogen (NBT/BCIP). This generates insoluble precipitates in the membrane in the positions where there has been hybridization. The results were analyzed automatically with the hybriSoft™ software (HSHS 2.2.0.R16 (HS12a)/HSHS IPL 1.0.2.R1000) (VITRO S.A.).

### 2.5. Extended Genotyping of the 12 HR Genotypes

Both HPV assays (Vitro HPV Screening and HPV Direct Flow CHIP) can be combined to perform extended genotyping, especially for the “other HR-HPV–positive” samples. In this study, all the HR-HPV–positive samples detected with the Vitro HPV Screening assay were directly genotyped with the HPV Direct Flow CHIP assay (Figure 1). The PCR products obtained from the Vitro HPV Screening PCR test were hybridized automatically into the HPV CHIP on the hybriSpot™12 PCRauto platform (Vitro S.A.), following the instructions and protocol recommended by the manufacturer. Although the HPV Direct Flow CHIP assay can identify up to 35 high- and low-risk genotypes, in this study only the 12 HR genotypes 31, 33, 35, 39, 45, 45, 51, 52, 56, 58, 59, 66, and 68 were considered for the analysis.

### 2.6. Statistical Analysis

Statistical analyses were performed at the Cancer Epidemiology Research Program of Catalan Institute of Oncology (ICO). The database with all laboratory results of the samples was created and provided by the Hospital del Mar to the ICO research staff for this purpose.

The clinical sensitivity and specificity of the Vitro HPV Screening test for CIN2+ and 95% confidence intervals (CI) were calculated using conventional contingency tables. The performance of the Vitro HPV Screening Real-Time PCR was compared with the cobas^®^ 4800 HPV Test by a non-inferiority test, following the criteria of Meijer [18]. According to guidelines, the gold standard to measure sensitivity must be a histologically-proven CIN2+, including CIN2-3, in situ adenocarcinoma and invasive cervical cancer. Considering the need of an HPV test to be used in large screening populations in different laboratories, the international expert guidelines emphasize the need for high reproducibility of an HPV test, which must be confirmed in the same laboratory as well as in different laboratories (intra- and inter-laboratory reproducibility) with a lower confidence bound not less than 87%. The sensitivity and specificity of the Vitro HPV Screening test should be at least 90% and 98%, respectively, compared to that of the cobas^®^ 4800 HPV Test. Observed agreement (%) and 95% CI were also calculated. For intra- and inter-laboratory reproducibility analyses, observed agreement (%) and a Cohen’s Kappa statistic with 95% CI were performed. Concordance by HPV type (HPV16, HPV18, and other HR HPV types) was also evaluated. Generally, a Kappa score between 0.8 and 1 are considered very good or almost perfect agreement, values between 0.61 and 0.8 are considered good agreement, values between 0.41 and 0.6 are considered moderate agreement, values between 0.21 and 0.4 are considered fair agreement, and values between 0 and 0.2 are considered slight agreement [34].

All statistical tests were two- tailed, and *p*-values below 0.05 were considered statistically significant. A non-inferiority test with a *p* of <0.05 means that the sensitivity or specificity of the Vitro HPV Screening Real-Time PCR kit is not significantly lower than that of the cobas^®^ 4800 HPV test.

Data analyses were carried out using R software version 4.2.1 (R Core Team (2022). R: A language and environment for statistical computing. R Foundation for Statistical Computing, Vienna, Austria. URL https://www.R-project.org/, accessed on 15 September 2022).

## 3. Results

### 3.1. Clinical Sensitivity and Specificity of Vitro HPV Screening Assay

HPV results with the cobas^®^ 4800 HPV Test and the Vitro HPV Screening assay for samples used in the sensitivity and specificity analyses are shown in Table 1. All the samples selected for sensitivity analysis (N = 60) had positive HPV results for both screening assays. However, in the samples selected for specificity analysis (N = 844), 109 (12.8%) were positive using the cobas^®^ HPV Test, and 108 (12.8%) were positive using the Vitro HPV Screening Real-Time PCR. Genotype distribution among HPV-positive women was the same for HPV16 (51.7% in the CIN2+ cases and 1.8% in the <CIN2 cases) and HPV18 (8.3% in the CIN2+ cases and 0.5% in the <CIN2 cases) for both assays. There was a discrepancy in one sample for 12 HR HPV genotypes (61.7% positivity using cobas^®^ HPV vs. 63.3% using Vitro HPV Screening in the CIN2+ cases, and 11.5% using cobas^®^ HPV vs. 11.4% using HPV Screening Real-Time PCR in the <CIN2 cases) (Table 1).

The Vitro HPV Screening test correctly identified the 60 samples with a histologic result of CIN2+, resulting in a clinical sensitivity for both assays of 100% (95% CI of 92.5–100). Specificity of the cobas^®^ HPV and Vitro HPV Screening tests for <CIN2 was 87.1% (95% CI: 84.6–89.2) and 87.2% (95% CI: 84.7–89.3), respectively (Table 1).

The relative sensitivity of the Vitro HPV Screening assay compared to the cobas^®^ HPV Test was 1 (Table 2) and was non-inferior to that of the cobas^®^ HPV (*p* = 0.0049).

The relative specificity of the Vitro HPV Screening test compared to the cobas^®^ HPV Test for the diagnosis of CIN2+ was 1 (95% CI: 0.97–1.00) (Table 2). The non-inferiority analysis showed that the clinical specificity of Vitro HPV Screening assay for a diagnosis of CIN2+ was not inferior to that of the cobas^®^ 4800 HPV (*p* < 0.001).

Regarding the analysis by HPV genotypes (Table 3), in the group of CIN2+ cases the agreement between Vitro HPV Screening and cobas^®^ 4800 HPV tests for HPV16 and HPV18 was 100% (95% CI: 92.5–100), with a Kappa value of 1 showing a perfect concordance. While for 12 HR HPV types was 98.3% (95% CI: 89.9–99.9), with a Kappa value of 0.96 (95% CI: 0.90–1.00).

In the cases <CIN2, the agreement between Vitro HPV Screening and cobas^®^ HPV tests for the HPV16 and HPV18 genotypes were 100% (95% CI: 99.4–100), with a Kappa value of 1. Concordance between both tests for the 12 HR HPV types was 99.9%, (95% CI: 99.2–100), with a Kappa value of 0.99 (95% CI: 0.98–1.00) (Table 3).

### 3.2. Intra-Laboratory Reproducibility and Inter-Laboratory Agreement

The intra-laboratory reproducibility analysis performed at the Hospital del Mar on two different days showed an agreement of 100% (95% CI: 99.2–100), with a Kappa value of 1 (Table 4). Both runs showed the same HPV positivity, with 46% of women being HPV-positive. Among the HPV-positive samples, genotype distribution was also the same for both runs: 11.4% of women positive for HPV16, 4.6% positive for HPV18, and 35.7% positive for high-risk HPV genotypes other than HPV16 and HPV18.

The inter-laboratory agreement was 97.3% (95% CI: 95.5–98.4), with a Kappa value of 0.95 (95% CI: 0.92–0.97) (Table 4). The samples analyzed at the Hospital del Mar showed HPV positivity of 46% compared to 43.5% in the samples analyzed in the ICO Laboratory. Among HPV-positive samples, genotype distribution showed 11.4% of women positive for HPV16 in the Hospital del Mar vs. 11.1% in the ICO, 4.6% positive for HPV18 in Hospital del Mar vs. 4.8% in the ICO, and 35.7% positive for high-risk HPV genotypes other than HPV16 and HPV18 in Hospital del Mar vs. 33.3% in the ICO. Among the 15 HPV-positive samples in the Hospital del Mar (some of those had a co-infection) that were negative in the ICO, 12 were positive for other HR HPVs, 2 HPV16-positive, and 1 HPV18-positive. However, most samples with discrepant results had high Ct values (approaching the 40-Ct, which is the manufacturer’s limit for determining an HPV-negative sample) suggesting low viral loads (Table 4).

### 3.3. Genotyping with the Vitro HPV Direct Flow Chip Assay

For all the samples in the study with HR HPV–positive results other than HPV16 and HPV18, the Vitro HPV Direct Flow CHIP assay was used to individually identify the 12 high-risk types: HPV31, HPV33, HPV35, HPV39, HPV45, HPV51, HPV52, HPV56, HPV58, HPV59, HPV66, or HPV68.

In those samples included in the sensitivity analysis (CIN2+) (Table 5), HPV31 was the most common genotype (N = 12 samples, 31.6%), followed by HPV52 (N = 9 samples, 23.7%) and HPV 58 (N = 6 samples, 15.8%). Regarding the least prevalent genotypes, only two samples (5.3%) were positive for HPV39, and no positive samples were found for the HPV59 genotype.

For the samples included in the specificity analysis (<CIN2 cases), HPV66 (N = 24 samples, 25%) was the most common genotype, followed by HPV52 (N = 22 samples, 22.9%) and HPV31 (N = 17 samples, 17.7%). The least prevalent genotypes were HPV33, HPV35 and HPV39 (Table 5).

In the samples used for the intra-laboratory and inter-laboratory reproducibility, although this analysis at the level of individual genotypes is not included in Meijer et al., guidelines [18], the concordance of specific HR genotypes was analyzed to complete the validation of the entire Vitro HPV detection technology.

The global intra-laboratory reproducibility was done at the Hospital del Mar, considering the results of HPV16, HPV18, and the pool of the 12 HR HPVs. However, the intra-laboratory reproducibility analysis of the extended genotyping of the 12 HR HPV samples was done retrospectively, several months later at the ICO laboratory, using the remainder of the samples. Of the 200 samples that were positive for the 12 HR genotypes, only 164 samples could be reanalyzed for individual genotyping, as there were 36 samples that did not have enough material. Two new DNA extractions and two HPV screening PCRs were performed with each sample on two different days; all positive samples for 12 HR HPV types were genotyped with the HPV Direct Flow CHIP test. The most prevalent genotype was HPV31 in both runs (18.3% and 19.5% in the first and in the second run, respectively), followed by HPV58 (16.5% in both runs), HPV56 (15.9% in the first run and 16.5% in the second), and HPV52 (15.2% in both runs). The least prevalent genotypes in both runs were HPV68 and HPV35 (Table 5).

Regarding the inter-laboratory agreement for individual genotypes, it was performed on the 187 samples that had tested positive in both laboratories for 12 HR HPV. Genotyping was not carried out on 13 samples that were positive for 12 HR HPV with Vitro HPV Screening test at Hospital del Mar but were negative for said test at the ICO. HPV31 was the most frequent genotype (21% in Hospital del Mar vs. 21.4% in ICO), followed by HPV58 (16% at Hospital del Mar vs. 15% in ICO) and HPV52 (15% in both sites). The least prevalent types were HPV68 (4% at Hospital del Mar vs. 3.7% in ICO) and HPV59 (5% and 5.9% respectively). In two samples from the Hospital del Mar, the individual genotype could not be determined (Table 5).

Table 6 shows the agreement and kappa values of individual HR HPV types other than HPV16 and HPV18 for the intra-laboratory reproducibility. There was high agreement for all the genotypes, ranging from 100% for HPV35, HPV39, and HPV59, 99.4% for HPV33, 98.8% for HPV31, HPV52, HPV58, and HPV66 to 98.2% for HPV51, HPV56, and HPV68. The kappa values ranged from 0.91 to 1 for all the genotypes, except for HPV68, which had a kappa value of 0.79, the lowest kappa among all genotypes.

The inter-laboratory agreement for individual genotypes showed, in general, a remarkably high agreement, ranging from 99.5% for HPV33, HPV35, HPV59, HPV66, and HPV68 to 96.3% for HPV 58. All Kappa values reached excellent agreement (kappa > 0.8). The highest Kappa value was obtained for HPV33 and HPV66 (Kappa of 0.97) and the lowest for HPV 58 (Kappa of 0.86) (Table 7).

## 4. Discussion

In this study, we have validated the clinical performance on ThinPrep-collected samples of the Vitro HPV Screening PCR assay vs. the comparator assay cobas^®^ HPV 4800, following the international guidelines for HPV DNA test requirements for primary cervical cancer screening in women 30 years and older [18]. The results of our validation study have shown that both sensitivity and specificity of the Vitro HPV Screening test are appropriate for cervical cancer screening, with almost perfect concordance with the reference test. High inter- and intra-laboratory reproducibility prove that it is a robust test for use as a primary screening test. The minor differences in the inter-laboratory reproducibility study (97.5% concordance) were mainly due to discrepancies between samples with Ct values close to the cutoff of the assay (fixed by the manufacturer at Ct = 40) and suggesting low viral loads, as is expected in comparative studies between PCR-based techniques [17].

This is the first article to validate this new assay based on the real-time multiplex PCR technique, targeting the conserved region of the L1 target gene of HPV genome. The Vitro HPV Screening PCR assay has already been CE marked under IVDR *(Regulation (EU) 2017/746)* for the detection of 14 HR HPV genotypes on clinical samples. The assay identifies HPV16 and HPV18 separately, and the other 12 HR genotypes, 31, 33, 35, 39, 45, 51, 52, 56, 58, 59, 66, and 68, as a pool. The amplification of human beta-globin gene in the same tube gives valuable information regarding the quality of the sample, and positive and negative controls are also included. In addition, the Vitro HPV Screening PCR assay has the entire process automated using a high-throughput modular platform, and complemented with individual genotyping, traceability, and automatic results interpretation that makes the Vitro HPV assay a firm candidate for screening of cervical cancer in population-based programs. Its clinical validation with self-sampling as compared to clinician-collected samples is currently being addressed in the VALHUDES framework.

It is important to emphasize that only tests detecting 14 high-risk HPVs should be validated and used in cervical cancer screening, avoiding the detection of low-risk HPVs that may lead to confusion and less specific tests. Moreover, appropriate sensitivity and specificity are fundamental for the impact of the new protocols on clinical results, to detect most of women at risk for high grade lesions and a minimum number of women with low or null risk. Our gold standard comparator test used for this validation was the cobas^®^ HPV 4800 Test instead of HC2 or GP5+/6+ PCR, which were recommended as gold standards at the time of publication of the validation guidelines [18]. The Vitro HPV Screening PCR assay is similar to the cobas^®^ HPV 4800, since it also detects HPV16 and HP18 individually and the 12 HR HPV types together. In 2021, Arbyn et al. published a systematic review of studies on HPV test validations following published guidelines (Meijer’s guidelines, Valgent) or cross-sectional studies [17]. At that time, some of the validation studies had already used other previously validated assays different from the originally recommended (HC2 or GP5+/6+ PCR), thus becoming new comparator HPV tests [35,36], among which cobas^®^ HPV 4800 was included. A need for updated guidelines including collecting devices and other targets is required since progress in the field is accelerating and new markers and devices are being developed and commercialized.

Although the evaluation of the detection of individual HPV genotypes is not currently required in the validation guidelines, and is therefore not usually reported, it was evaluated and included in this study. Most screening programs’ criteria assess clinical performance on HPV16 and HPV18 to be managed differently than the remaining 12 HR genotypes, and a positive result for any of the 12 HR HPV genotypes has traditionally been managed by cytology triage [37]. However, there are new approaches for risk stratification and clinical management using individual genotyping information [38,39,40]. The clinical utility of HPV genotyping in risk determination during cervical cancer screening has been extensively described based upon evidence that the HPV genotypes have different oncogenic potentials [41,42]. In this sense the validation of HPV tests for genotyping is needed and there are studies with this objective [43]. Our study has included the genotyping of all the HPV-positive samples. The objective was not a validation study itself, but the results can be the basis for future comparisons with validated tests on studies like Valgent, an international initiative [25]. We found HPV31 to be the most frequent type among samples from the CIN2+ cases, followed by HPV52 and HPV58, with the HPV types 68, 39, and 59 the least detected genotypes in this group. These are new data on our screening target population that may have clinical impact in the future when vaccination will be more extensive among the screening population. Among samples from <CIN2 cases, HPV66 was the most common genotype, followed by HPV52 and HPV31. The clinical significance of these findings cannot be assessed in this study, however, these data are consistent with previously published data [41,44], which suggest that HPV types 16, 18, 31, 33, 52 and 58 represent high-to-moderate risk of cervical disease, while HPV genotypes like 39, 59 or 68 represent low risk in cervical cancer, and support the strategy of using genotyping in a screening algorithm in order to stratify HPV-positive women for follow-up and avoid overtreatments. On the other hand, HPV66 is less associated with high-grade lesions and could allow a more conservative approach.

Reproducibility among the individual genotypes showed excellent concordance for all 12 genotypes for both intra-laboratory reproducibility (kappa ranging from 1 to 0.79) and inter-laboratory agreement (kappa ranging from 0.97 to 0.86). The lowest kappa was for HPV68, which was also one of the least frequent genotypes in the positive samples (4.9%). Nevertheless, the concordance for this genotype was 98.2% in the intra-laboratory analysis and therefore its clinical implication in the overall concordance is minimal. Individual genotyping may become the best option in the future when the cohort of vaccinated women come to the age of HPV-based screening.

The Vitro HPV Screening assay, complemented with extended genotyping of the 12 HR HPV by the HPV Direct Flow CHIP test, has the capacity to obtain genotyping information on HPV-positive samples without further analysis in an integrated, fully automated workflow, and represents a unique solution for primary cervical cancer screening. Genotyping with HPV Direct Flow CHIP requires further validation within international trials; however, its easy handling can facilitate its introduction as a clinical tool to establish individual risk stratification of HPV-positive women in the near future.

## 5. Conclusions

In conclusion, our data shows that the Vitro HPV Screening assay presents non-inferior accuracy to the comparator cobas^®^ HPV 4800 assay with respect to clinical sensitivity and specificity for cervical cancer screening on ThinPrep-collected cervical samples in a population of women aged 30 years and older. Inter-laboratory agreement and intra-laboratory reproducibility also fulfill the international validation criteria. The results of this validation study support the Vitro HPV Screening assay as a suitable candidate for future primary HPV screening.

## Figures and Tables

**Figure 1 cancers-16-01322-f001:**
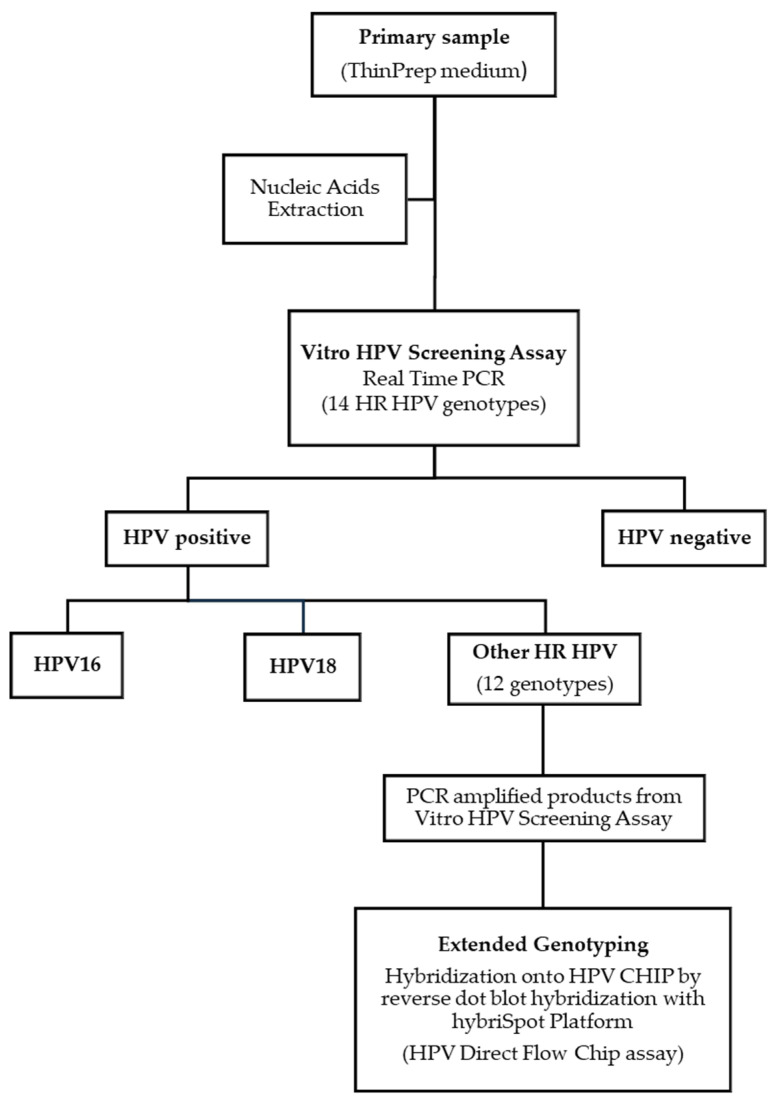
Workflow for HR HPV detection and extended HPV genotyping through the Vitro HPV Screening test.

**Table 1 cancers-16-01322-t001:** HPV results distribution and sensitivity and specificity of the Vitro HPV Screening and cobas^®^ 4800 HPV Assays.

	CIN2+ Cases	Clinical Sensitivity % (95% CI)	<CIN2 Cases	Clinical Specificity % (95% CI)
	N (%)		N (%)	
Vitro HPV Screening		100		87.2
HPV-Negative	0 (0)	(92.5–100)	736 (87.2)	(84.7–89.3)
HPV-Positive	60 (100)		108 (12.8)	
HPV16	31 (51.7)		15 (1.8)	
HPV18	5 (8.3)		4 (0.5)	
12 HR HPVs	38 (63.3)		96 (11.4)	
cobas^®^ 4800 HPV		100		87.1
HPV-Negative	0 (0)	(92.5–100)	735 (87.1)	(84.6–89.2)
HPV-Positive	60 (100)		109 (12.9)	
HPV16	31 (51.7)		15 (1.8)	
HPV18	5 (8.3)		4 (0.5)	
12 HR HPVs	37 (61.7)		97 (11.5)	
Total	60		844	

CI = confidence interval. 12 HR HPVs: HPV-positive for high-risk types other than HPV16 and HPV18. In the event of co-infection with several genotypes, one sample is counted for each different HPV type, hence the total N by genotypes is greater than the total N of positives, and the total % by genotypes is greater than 100%.

**Table 2 cancers-16-01322-t002:** Clinical performance of Vitro HPV Screening assay compared to cobas^®^ 4800 HPV assay.

		Cobas^®^ 4800 HPV				Non-Inferiority Test ^b^
		HPV- Positive ^a^	HPV- Negative	Total		
CIN2+ Cases					Relative Sensitivity (95% CI)	
Vitro HPV Screening	HPV- Positive ^a^	60	0	60	1.00 (-)	*p* = 0.0049
	HPV- Negative	0	0	0		
	Total	60	0	60		
<CIN2 cases					Relative Specificity (95% CI)	
Vitro HPV Screening	HPV- Positive ^a^	108	0	108	1.00 (0.97–1.00)	*p* < 0.001
	HPV- Negative	1	735	736		
	Total	109	735	844		

CI = confidence interval. ^a^ Positive for any of the 14 HR HPV genotypes detected with both assays. ^b^ A non-inferiority test with a *p* of <0.05 means that the sensitivity or specificity of the Vitro HPV Screening Real-Time PCR assay is not significantly lower than that of the cobas^®^ 4800 HPV.

**Table 3 cancers-16-01322-t003:** Concordance and Kappa values of the Vitro HPV Screening test for HPV16, HPV18 and 12 high-risk genotypes relative to the cobas^®^ 4800 HPV Test for CIN2+ and <CIN2 results among women included in the sensitivity and specificity analysis, respectively.

	HPV Results				Agreement (%)	95% CI	Kappa Value	95% CI
	P1/P2	N1/P2	P1/N2	N1/N2				
Sensitivity analysis (CIN2+ cases)								
HPV16	31	0	0	29	100	92.5–100	1	- ^c^
HPV18	5	0	0	55	100	92.5–100	1	- ^c^
12 HR HPVs	37	1 ^a^	0	22	98.3	89.9–99.9	0.96	0.90–1.00
Specificity analysis (<CIN2 cases)								
HPV16	15	0	0	829	100	99.4–100	1	- ^c^
HPV18	4	0	0	840	100	99.4–100	1	- ^c^
12 HR HPVs	96	0	1 ^b^	747	99.9	99.2–100	0.99	0.98–1.00

P = HPV result positive; N = HPV result negative; 1 = cobas^®^ 4800 HPV Test; 2 = Vitro HPV Screening test, CI = confidence interval. 12 HR HPVs: HPV positive for high-risk types other than HPV16 and HPV18. ^a^ Discordant HR HPV, positive for HPV68 with Vitro HPV Screening test (Ct value: 27.06). ^b^ Discordant HR HPV, positive for HR HPV with cobas^®^ HPV (Ct: 35.2).^c^ A Kappa value of 1 doesn’t allow the Kappa confidence interval to be calculated, returning an error when the corresponding formula is used.

**Table 4 cancers-16-01322-t004:** Intra-laboratory reproducibility and inter-laboratory agreement of the Vitro HPV Screening assay.

		Hospital Del Mar Run 1			Agreement % (95% CI)	Kappa Value (95% CI)
Hospital del Mar Run 2		HPV-Positive	HPV-Negative	Total (%)		
	HPV-Positive	258	0	258 (46%)	100	1
	HPV-Negative	0	303	303 (54%)	(99.2–100)	(-) ^b^
	Total (%)	258 (46%)	303 (54%)	561 (100%)		
ICO Run 1		HPV-Positive	HPV-Negative	Total (%)		
	HPV-Positive	243	0	243 (43.3%)	97.3	0.95
	HPV-Negative	15 ^a^	303	318 (56.7%)	(95.5–98.4)	(0.92–0.97)
	Total (%)	258 (46%)	303 (54%)	561 (100%)		

ICO *=* Catalan Institute of Oncology; CI = confidence interval. ^a^ Discrepant HPV results that were HPV-positive with the Vitro HPV Screening test when analyzed in Hospital del Mar: 12 samples were positive for 12 HR HPV (Ct values: 39.2, 38.8, 36.1, 36.8, 37.3, 38.8, 29.1, 28.0, 30.5, 32.5, 27.3, 25.2), 2 samples were positive for HPV16 (Ct values: 37.4, 30.4) and 1 sample was positive for HPV18 (Ct: 37.9). ^b^ A Kappa value of 1 doesn’t allow the Kappa confidence interval to be calculated, returning an error when the corresponding formula is used.

**Table 5 cancers-16-01322-t005:** Distribution of high-risk HPV genotypes other than HPV16 and HPV18 identified with HPV Direct Flow CHIP test, among women who reported “other HR HPVs”-positive results with Vitro HPV Screening assay.

HPV Direct Flow CHIP Assay	CIN2+ Cases	<CIN2 Cases	Intra-Laboratory Reproducibility (ICO)		Inter-Laboratory Agreement	
			First Run	Second Run	Hospital Del Mar ^c^	ICO
	N (%)	N (%)	N (%)	N (%)	N (%)	N (%)
VPH31	12 (31.6)	17 (17.7)	30 (18.3)	32 (19.5)	42 (21)	40 (21.4)
VPH33	5 (13.2)	6 (6.3)	14 (8.5)	13 (7.9)	18 (9)	18 (9.6)
VPH35	4 (10.5)	6 (6.3)	11 (6.7)	11 (6.7)	14 (7)	15 (8)
VPH39	2 (5.3)	6 (6.3)	23 (14)	23 (14)	25 (12.5)	26 (13.9)
VPH45	4 10.5)	9 (9.4)	20 (12.2)	19 (11.6)	24 (12)	20 (10.7)
VPH51	4 (10.5)	13 (13.5)	20 (12.2)	19 (11.6)	21 (10.5)	17 (9.1)
VPH52	9 (23.7)	22 (22.9)	25 (15.2)	25 (15.2)	30 (15)	28 (15)
VPH56	4 (10.5)	9 (9.4)	26 (15.9)	27 (16.5)	29 (14.5)	26 (13.9)
VPH58	6 (15.8)	9 (9.4)	27 (16.5)	27 (16.5)	32 (16)	28 (15)
VPH59	0 (0)	14 (14.6)	12 (7.3)	12 (7.3)	10 (5)	11 (5.9)
VPH66	4 (10.5)	24 (25)	19 (11.6)	17 (10.4)	19 (9.5)	19 (10.2)
VPH68	3 (7.9)	8 (8.3)	8 (4.9)	7 (4.3)	8 (4)	7 (3.7)
Total Samples	38 ^a^	96	164	164	200	187
Samples with insufficient material ^b^			36			

ICO *=* Catalan Institute of Oncology. ^a^ In one sample the individual genotype(s) could not be determined. ^b^ There was insufficient sample volume for processing with HPV Direct Flow CHIP. ^c^ The genotypes could not be identified in two samples from the Hospital del Mar. In the event of co-infection with several genotypes, one sample iss counted for each different HPV type, hence the total N by genotypes is greater than the total N of positives, and the total % by genotypes is greater than 100%.

**Table 6 cancers-16-01322-t006:** Agreement and kappa values for the intra-laboratory reproducibility of HR HPV types other than HPV16 and HPV18, genotyped with the HPV Direct Flow CHIP assay.

HPV Direct Flow CHIP Assay	Individual Genotype Findings				Agreement (%)	95% CI	Kappa	95% CI
HR HPV Type	P1/P2	N1/P2	P1/N2	N1/N2				
VPH31	30	2	0	132	98.8	97.1–100	0.96	0.90–1.00
VPH33	13	0	1	150	99.4	98.2–100	0.96	0.88–1.00
VPH35	11	0	0	153	100	100–100	1.00	1.00–1.00
VPH39	23	0	0	141	100	100–100	1.00	1.00–1.00
VPH45	19	0	1	144	99.4	98.2–100	0.97	0.91–1.00
VPH51	18	1	2	143	98.2	96.1–100	0.91	0.81–1.00
VPH52	24	1	1	138	98.8	97.1–100	0.95	0.88–1.00
VPH56	25	2	1	136	98.2	96.1–100	0.93	0.86–1.00
VPH58	26	1	1	136	98.8	97.1–100	0.96	0.89–1.00
VPH59	12	0	0	152	100	100–100	1.00	1.00–1.00
VPH66	17	0	2	145	98.8	97.1–100	0.94	0.85–1.00
VPH68	6	1	2	155	98.2	96.1–100	0.79	0.56–1.00

The genotyping was performed at the Catalan Institute of Oncology. CI = confidence interval, P = HPV result positive; N = HPV result negative; 1 = run 1; 2 = run 2.

**Table 7 cancers-16-01322-t007:** Inter-laboratory agreement and Kappa values of individual HR HPV types other than HPV16 and HPV18, genotyped with the HPV Direct Flow CHIP assay.

HPV Direct Flow CHIP Assay	Individual Genotype Findings				Agreement (%)	95% CI	Kappa	95% CI
HR HPV Type	P1/P2	N1/P2	P1/N2	N1/N2				
VPH31	38	2	1	146	98.4	95.0–99.6	0.95	0.90–1.00
VPH33	17	1	0	169	99.5	96.6–100	0.97	0.91–1.00
VPH35	14	1	0	172	99.5	96.6–100	0.96	0.89–1.00
VPH39	24	2	1	160	98.4	95.0–99.6	0.93	0.86–1.00
VPH45	19	1	1	166	98.9	95.7–99.8	0.94	0.87–1.00
VPH51	17	0	4	166	97.9	94.3–99.3	0.88	0.77–0.99
VPH52	26	2	4	155	96.8	92.8–98.7	0.88	0.78–0.97
VPH56	25	1	3	158	97.9	94.3–99.3	0.91	0.83–0.99
VPH58	25	3	4	155	96.3	92.1–98.3	0.86	0.75–0.96
VPH59	10	1	0	176	99.5	96.6–100	0.95	0.85–1.00
VPH66	18	1	0	168	99.5	96.6–100	0.97	0.91–1.00
VPH68	7	0	1	179	99.5	96.6–100	0.93	0.80–1.00

The genotyping was performed at Hospital del Mar and at the Catalan Institute of Oncology (ICO). CI = confidence interval. P = HPV result positive; N = HPV result negative; 1 = Hospital del Mar 1; 2 = ICO.

## Data Availability

The data presented in this study are available upon request from the corresponding author.

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
