# Peer review of "Clinical Validation of the Vitro HPV Screening Assay for Its Use in Primary Cervical Cancer Screening"

_cancers, 2024, doi:10.3390/cancers16071322_

Round 1

Reviewer 1 Report

Comments and Suggestions for Authors

Human papillomaviruses (HPV) infection is known as the main cause of cervical cancer and precursor lesions. Cervical cancer screening tests are focused on high-risk human papillomavirus (HR HPV) detection via different techniques (hybridization, signal amplification, DNA or RNA, targeted gene, or other). However, there is no one ideal test yet for detection of various types of HR HPV. In this study, authors demonstrated that the Vitro HPV Screening assay targeting 14 oncogenic genotypes (HPV16, HPV18 and other 12 HR HPV) showed a high intra- and inter-laboratory reproducibility, also among the individual genotypes. The Vitro HPV Screening assay is valid for cervical cancer screening. The study was well designed and performed. The report was written clearly except a few concerns listed below.

1.       The cobas® 4800 HPV Test was mentioned as a gold standard assay. When it appears first at line 93, briefly describe how it works.

2.       Residual cervical specimens from Thinprep were used in the Vitro HPV Screening assay. What is the minimal amount (ng) of the HPV and cellular DNA that is required per PCR reaction?

3.       In the real-time PCR for detecting 14 HR-HPVs, positivity of each HPV16, HPV18 or 12 HR-HPVs was defined by ct value-cutoff of 40 in this validation. How was the ct value determined for detecting of various HR-HPVs?

4.       In this validation study, the assay specifically detects 14 HR HPVs in separated fluorescent channels. In the multiplex fluorescent PCR, whether one, 3, or 14 primer pairs and different fluorescent probes were used to identify HPV types 16, 18, and 12 HR HPV was not clear. Were the PCR amplicons of HPV16, HPV18, and other 12 HR-HPV types (31, 33, 35, 39, 45, 51, 52, 56, 58, 59, 66 and 68) distinguishable under the same PCR condition? How were these PCR products validated for their specificity?

Author Response

MDPI Tongzhou

Cancers

22nd March 2024

Dear first reviewer,

Thank you very much for your comments. Below you will find point-by-point responses to your comments.

Please, also, find attached a thoroughly revised version of our original article entitled: “Clinical validation of the Vitro HPV screening assay for its use in primary cervical cancer screening” (manuscript ID: cancers-2916730). To facilitate the review process, the changes made have been highlighted in the revised manuscript using underlining. This revised document incorporates all the suggestions made by the three reviewers.

We thank you for your time and consideration of our work.

Yours sincerely,

Belen Lloveras

Hospital del Mar-IMIM

Passeig Maritim 25-29

Barcelona, Spain 08003

+34932483521

Reviewer: 1

Comments to the Author

Human papillomaviruses (HPV) infection is known as the main cause of cervical cancer and precursor lesions. Cervical cancer screening tests are focused on high-risk human papillomavirus (HR HPV) detection via different techniques (hybridization, signal amplification, DNA or RNA, targeted gene, or other). However, there is no one ideal test yet for detection of various types of HR HPV. In this study, authors demonstrated that the Vitro HPV Screening assay targeting 14 oncogenic genotypes (HPV16, HPV18 and other 12 HR HPV) showed a high intra- and inter-laboratory reproducibility, also among the individual genotypes. The Vitro HPV Screening assay is valid for cervical cancer screening. The study was well designed and performed. The report was written clearly except a few concerns listed below.

  1. The cobas® 4800 HPV Test was mentioned as a gold standard assay. When it appears first at line 93, briefly describe how it works.

Authors’ Response:

We thank the reviewer for this comment. We have added the following text to explain how cobas® 4800 HPV Test works (page 2, line 93-96):

 “…..a real-time PCR test that detects 14 HR-HPV in three different fluorescent channels, targeting the L1 region of the HPV genome. HPV16 and HPV18 are detected separately and the other 12 HR-HPV types (31, 33, 35, 39, 45, 51, 52, 56, 58, 59, 66 and 68) are detected as a pool by a cocktail of probes in a separated channel”.

  1. Residual cervical specimens from Thinprep were used in the Vitro HPV Screening assay. What is the minimal amount (ng) of the HPV and cellular DNA that is required per PCR reaction?

Authors’ Response:

We thank the reviewer for this comment. The extraction of DNA was performed with the MAIS automated system (Vitro S.A.) and the RNA/DNA Pathogen Extraction kit (Vitro S. A.). The automated process starts from 92 µl of clinical Thinprep sample, the DNA is eluted in 60 µl and 8 µl are used as template for PCR amplification (20 ul final reaction volume). The minimal amount (ng) of cellular DNA for the test was stablished at 10ng/reaction. The LoD of the Vitro HPV Screening test for HPV detection was stablished at 10 GE/reaction for the 14 HR HPV genotypes.

We have incorporated in the manuscript the requested information as follows (page 4, line 183): “The minimal amount (ng) of cellular DNA for the test was stablished at 10ng/reaction”.

  1. In the real-time PCR for detecting 14 HR-HPVs, positivity of each HPV16, HPV18 or 12 HR-HPVs was defined by ct value-cutoff of 40 in this validation. How was the ct value determined for detecting of various HR-HPVs?

Authors’ Response:

We appreciate the reviewer’s comment. The cutoff for the Vitro HPV Screening assay has been set at a Ct ≤ 40. Thus, any sample presenting a Ct value ≤ 40 will be considered positive. This value has been established considering the Cts obtained in assays using synthetic sequences of each of the targets included in the kit as template at their LoD. In these assays, the Cts obtained for each of the targets under these conditions were found to be less than or equal to 40. Likewise, to establish this cut-off value, the Cts obtained with the set of clinical samples that have been analyzed in the clinical validation studies compared to the reference kit have been considered. In all cases the Ct of the amplification curve was below 40 for all the positive samples. No false positive results were obtained for negative clinical samples, considering this Ct value.

We have incorporated in the manuscript the requested information as follows (page 4, line 177-178): “The Ct value was defined following sensitivity and specificity assays developed by the manufacturer”. 

  1. In this validation study, the assay specifically detects 14 HR HPVs in separated fluorescent channels. In the multiplex fluorescent PCR, whether one, 3, or 14 primer pairs and different fluorescent probes were used to identify HPV types 16, 18, and 12 HR HPV was not clear. Were the PCR amplicons of HPV16, HPV18, and other 12 HR-HPV types (31, 33, 35, 39, 45, 51, 52, 56, 58, 59, 66 and 68) distinguishable under the same PCR condition? How were these PCR products validated for their specificity?

Authors’ Response:

We thank the reviewer for raising this observation. The assay is a multiplex real time PCR containing a pool of primers targeting the L1 region of HPV genome (GP5+/GP6+). It detects 14 HR genotypes in 3 separated fluorescent channels by using 14 genotype specific fluorescent probes. HPV 16 amplicons are detected by a ROX-labelled probe, HPV 18 amplicons are detected by a Cy5-labelled probe, and the 12 HR (31, 33, 35, 39, 45, 51,52, 56, 58, 59, 66, 68) amplified signals are detected by genotype-specific probes labelled with the same fluorescent dye (FAM). HPV 16 and HPV 18 can be distinguished separately, and the 12 HR genotypes are detected as a pool.

The specificity of each specific probe for detecting each of the 12 HR HPV genotypes was verified by the manufacturer analytically by testing separately each specific probe with its corresponding HPV genotype.

To clarify this point in the manuscript we have added the following sentence in page 4, line 168-169):” The assay is a multiplex real time PCR containing a pool of primers targeting the L1 region of HPV genome and 14 genotype specific fluorescent probes.”

Reviewer 2 Report

Comments and Suggestions for Authors

The authors  should provide a workflow, and after that it could be accepted.

Author Response

MDPI Tongzhou

Cancers

22nd March 2024

Dear second reviewer,

Thank you very much for your comments. Below you will find point-by-point responses to your comments.

Please, also, find attached a thoroughly revised version of our original article entitled: “Clinical validation of the Vitro HPV screening assay for its use in primary cervical cancer screening” (manuscript ID: cancers-2916730). To facilitate the review process, the changes made have been highlighted in the revised manuscript using underlining.This revised document incorporates all the suggestions made by the three reviewers.

We thank you for your time and consideration of our work.

Yours sincerely,

Belen Lloveras

Hospital del Mar-IMIM

Passeig Maritim 25-29

Barcelona, Spain 08003

+34932483521

Reviewer: 2

The authors should provide a workflow, and after that it could be accepted.

Authors’ Response:

We thank the reviewer for this comment. With "workflow" unspecified, we assume it refers to the Vitro workflow. If it's otherwise or refers to another workflow, please let us know. A figure with the workflow for HR HPV detection and extended HPV genotyping through the Vitro HPV screening test has been included in the manuscript (page 5 line 211) and labeled as Figure 1. Workflow for HR HPV detection and extended HPV genotyping. Figure 1 has been included in the text in page 4 line 168 and page 5, line 205).  

Reviewer 3 Report

Comments and Suggestions for Authors

Performance of the described conduct of the investigation was outstanding, recognizing the need for evaluating the sensitivity and specificity of HPV detection tests (Vitro HPV), and using the standard comparator method (cobas HPV 4800). This necessary analysis is important to provide the uniform cervical cancer screening methodology to be used by clinicians worldwide. The presentation of the rationale, the laboratory assay performance, and the detailed results, is appreciated by this reviewer.

There are no errors found in this excellent report.

Author Response

MDPI Tongzhou

Cancers

22nd March 2024

Dear third reviewer,

Thank you very much for your comments. Below you will find point-by-point responses to your comments.

Please, also, find attached a thoroughly revised version of our original article entitled: “Clinical validation of the Vitro HPV screening assay for its use in primary cervical cancer screening” (manuscript ID: cancers-2916730). To facilitate the review process, the changes made have been highlighted in the revised manuscript using underlining. This revised document incorporates all the suggestions made by the three reviewers.

We thank you for your time and consideration of our work.

Yours sincerely,

Belen Lloveras

Hospital del Mar-IMIM

Passeig Maritim 25-29

Barcelona, Spain 08003

+34932483521

Reviewer: 3

Performance of the described conduct of the investigation was outstanding, recognizing the need for evaluating the sensitivity and specificity of HPV detection tests (Vitro HPV), and using the standard comparator method (cobas HPV 4800). This necessary analysis is important to provide the uniform cervical cancer screening methodology to be used by clinicians worldwide. The presentation of the rationale, the laboratory assay performance, and the detailed results, is appreciated by this reviewer.

There are no errors found in this excellent report.

Authors’ Response:

We thank the reviewer for taking the time in reviewing our manuscript and for the kind comment on our work.